# Speech-induced suppression and vocal feedback sensitivity in human cortex

Muge Ozker[1,2]*, Leyao Yu[1,3], Patricia Dugan[1], Werner Doyle[4], Daniel Friedman[1], Orrin Devinsky[1], Adeen Flinker[1,3]

[1]Neurology Department, New York University, New York, United States; [2]Max Planck Institute for Psycholinguistics, Nijmegen, Netherlands; [3]Biomedical Engineering Department, New York University, New York, United States; [4]Neurosurgery Department, New York University, New York, United States

*For correspondence:
mozker@gmail.com

Competing interest: The authors declare that no competing interests exist.

**Abstract** Across the animal kingdom, neural responses in the auditory cortex are suppressed during vocalization, and humans are no exception. A common hypothesis is that suppression increases sensitivity to auditory feedback, enabling the detection of vocalization errors. This hypothesis has been previously confirmed in non-human primates, however a direct link between auditory suppression and sensitivity in human speech monitoring remains elusive. To address this issue, we obtained intracranial electroencephalography (iEEG) recordings from 35 neurosurgical participants during speech production. We first characterized the detailed topography of auditory suppression, which varied across superior temporal gyrus (STG). Next, we performed a delayed auditory feedback (DAF) task to determine whether the suppressed sites were also sensitive to auditory feedback alterations. Indeed, overlapping sites showed enhanced responses to feedback, indicating sensitivity. Importantly, there was a strong correlation between the degree of auditory suppression and feedback sensitivity, suggesting suppression might be a key mechanism that underlies speech monitoring. Further, we found that when participants produced speech with simultaneous auditory feedback, posterior STG was selectively activated if participants were engaged in a DAF paradigm, suggesting that increased attentional load can modulate auditory feedback sensitivity.

## eLife assessment

The manuscript describes human intracranial neural recordings in the auditory cortex during speech production, showing that the effects of delayed auditory feedback correlate with the degree of underlying speech-induced suppression. This is an **important** finding, as previous work has suggested that speech suppression and feedback sensitivity often do not co-localize and may be distinct processes, in contrast with findings in non-human primates where there is a strong correlation. The strength of the evidence is **convincing**, with appropriate experimental methods, data, and analysis.

## Introduction

A major question in neuroscience is how do animals distinguish between stimuli originating from the environment and those produced by their own actions. Sensorimotor circuits share a common mechanism across the animal kingdom in which sensory responses to self-generated motor actions are suppressed. It is commonly hypothesized that suppressing responses to predicted self-generated stimuli increases sensitivity of the sensory system to external stimuli (*Poulet and Hedwig, 2002*; *Poulet and Hedwig, 2006*; *Crapse and Sommer, 2008*; *Schneider and Mooney, 2018*). Furthermore, it enables detection and correction of motor errors by providing a template of the predicted

**eLife digest** The brain lowers its response to inputs we generate ourselves, such as moving or speaking. Essentially, our brain 'knows' what will happen next when we carry out these actions, and therefore does not need to react as strongly as it would to unexpected events. This is why we cannot tickle ourselves, and why the brain does not react as much to our own voice as it does to someone else's. Quieting down the brain's response also allows us to focus on things that are new or important without getting distracted by our own movements or sounds.

Studies in non-human primates showed that neurons in the auditory cortex (the region of the brain responsible for processing sound) displayed suppressed levels of activity when the animals made sounds. Interestingly, when the primates heard an altered version of their own voice, many of these same neurons became more active. But it was unclear whether this also happens in humans.

To investigate, Ozker et al. used a technique called electrocorticography to record neural activity in different regions of the human brain while participants spoke. The results showed that most areas of the brain involved in auditory processing showed suppressed activity when individuals were speaking. However, when people heard an altered version of their own voice which had an unexpected delay, those same areas displayed increased activity. In addition, Ozker et al. found that the higher the level of suppression in the auditory cortex, the more sensitive these areas were to changes in a person's speech.

These findings suggest that suppressing the brain's response to self-generated speech may help in detecting errors during speech production. Speech deficits are common in various neurological disorders, such as stuttering, Parkinson's disease, and aphasia. Ozker et al. hypothesize that these deficits may arise because individuals fail to suppress activity in auditory regions of the brain, causing a struggle when detecting and correcting errors in their own speech. However, further experiments are needed to test this theory.

sensory outcome to compare with the actual sensory outcome. In the domain of speech, this mechanism is described in models which suggest that neural responses in the auditory cortex are suppressed during speech production. When there is a mismatch between the predicted auditory outcome and the actual auditory feedback, responses in the auditory regions are enhanced to encode the mismatch and inform vocal-motor regions to correct vocalization (*Hickok et al., 2011*; *Houde and Nagarajan, 2011*; *Tourville and Guenther, 2011*).

A common experimental strategy to generate mismatch between the predicted auditory outcome and the actual auditory feedback is to perturb auditory feedback during speech production. Auditory feedback perturbations are usually applied either by delaying auditory feedback (DAF), which disrupts speech fluency (*Lee, 1950*; *Fairbanks, 1955*; *Stuart et al., 2002*), or by shifting voice pitch and formants, which result in compensatory vocal changes in the opposite direction of the shift (*Houde and Jordan, 1998*; *Jones and Munhall, 2000*; *Niziolek and Guenther, 2013*). Numerous electrophysiological and neuroimaging studies investigated neural responses during speech production both in the absence and presence of auditory feedback perturbations. In support of speech production models, these studies have repeatedly reported suppressed responses in auditory cortex during speaking compared with passive listening to speech (*Numminen et al., 1999*; *Wise et al., 1999*; *Curio et al., 2000*, *Houde et al., 2002*; *Christoffels et al., 2007*, *Ford et al., 2010*, *Niziolek et al., 2013*), as well as enhanced responses when auditory feedback was perturbed indicating sensitivity to auditory feedback (*Tourville et al., 2008*; *Behroozmand et al., 2009*; *Chang et al., 2013*; *Greenlee et al., 2013*, *Kort et al., 2014*; *Behroozmand et al., 2015*; *Ozker et al., 2022*). However, it is not clear whether the same or distinct neural populations in the auditory cortex show speech-induced suppression and sensitivity to auditory feedback.

While auditory responses are largely suppressed during speech production, detailed investigations using neurosurgical recordings revealed that the degree of suppression was variable across cortical sites, and auditory cortex also exhibited non-suppressed and enhanced responses (albeit less common) (*Creutzfeldt and Ojemann, 1989*; *Flinker et al., 2010*; *Greenlee et al., 2011*), mirroring results from non-human primate studies using single-unit recordings (*Eliades and Wang, 2003*; *Eliades and Wang, 2008*). In the same non-human primate study, it was reported that neurons that

were suppressed during vocalization showed increased activity when auditory feedback was perturbed (*Eliades and Wang, 2008*). Based on this finding, we predicted that if speech-induced suppression enables detection and correction of speech errors, suppressed auditory sites should be sensitive to auditory feedback, thus exhibit enhanced neural responses to feedback perturbations. Alternatively, if suppression and speech monitoring are unrelated processes, then suppressed sites should be distinct from the ones that are sensitive to auditory feedback.

The level of attention during speech monitoring can vary depending on the speech task. During normal speech production, speech monitoring does not require a conscious effort, however it is a controlled, attentional process during an auditory feedback perturbation task (*Hashimoto and Sakai, 2003*). It is well known that selective attention enhances auditory responses and improves speech perception under noisy listening conditions or when multiple speech streams are present (*Mesgarani and Chang, 2012*, *Zion Golumbic et al., 2013*). We predicted that increased attention to auditory feedback under adverse speaking conditions, such as during an auditory feedback perturbation task, should increase feedback sensitivity and elicit larger responses in the auditory cortex compared to normal speech production.

To summarize, in this study we aimed to test the hypothesis that speech-induced suppression increases sensitivity to auditory feedback in human neurophysiological recordings. We predicted that auditory sites showing speech-induced suppression would elicit enhanced responses to auditory feedback perturbations. Further, we aimed to investigate the role of attention in auditory feedback sensitivity by comparing auditory responses during an auditory feedback perturbation task compared with normal speech production.

To address these aims, we used intracranial electroencephalography (iEEG) recordings in neurosurgical participants, which offers a level of spatial detail and temporal precision that would not be possible to achieve using non-invasive techniques. We first identified the sites that show auditory suppression during speech production, and then employed a DAF paradigm to test whether the same sites show sensitivity to perturbed feedback. Our results revealed that overlapping sites in the superior temporal gyrus (STG) exhibited both speech-induced auditory suppression and sensitivity to auditory feedback with a strong correlation between the two measures, supporting the hypothesis that auditory suppression predicts sensitivity to speech errors in humans. Further, we showed that auditory responses in the posterior STG are enhanced in a DAF task compared to normal speech production, even for trials in which participants receive simultaneous auditory feedback (no-delay condition). This result suggests that increased attention during an auditory feedback perturbation task can modulate auditory feedback sensitivity and posterior STG is a critical region for this attentional modulation.

## Results

In order to assess cortical responses during perception and production of speech, and quantify speech-induced auditory suppression, participants (n=35) performed an auditory word repetition (AWR) task. We examined the response patterns in seven different cortical regions including STG, middle temporal gyrus (MTG), supramarginal gyrus (SMG), inferior frontal gyrus (IFG), middle frontal gyrus (MFG), precentral gyrus (preCG), and postcentral gyrus (postCG) (*Figure 1A*). As an index of the neural response, we used the high gamma broadband signal (70–150 Hz, *see Materials and methods*), which correlates with the spiking activity of the underlying neuronal population (*Mukamel et al., 2005*; *Crone et al., 2006*; *Cardin et al., 2009*; *Ray and Maunsell, 2011*; *Lachaux et al., 2012*).

We analyzed the responses in two different time windows: during passive listening of the auditory stimulus (0–500 ms after stimulus onset) and during speaking when participants repeated the perceived auditory stimulus (0–500 ms after articulation onset). Average responses were larger during passive listening in STG (average % signal change ± SEM; Listen: 62.1±0.6, Speak: 29.8±0.4), MTG (32.7±0.9, 22.3±0.9), and SMG (27.4±0.8, 25.8±0.7) compared with speaking. Conversely, responses were larger during speaking in IFG (29.2±1.3, 31.2±1.3), MFG (28.3±1.6, 31.4±1.3), preCG (27.4±0.4, 37±0.5), and postCG (26±0.4, 42±0.5). These results suggested that auditory regions responded more strongly during passive listening compared to speaking, verifying previous reports of neural response suppression to self-generated speech in auditory cortex (*Figure 1B–D*).

In the AWR task, participants heard the same auditory stimulus twice in each trial, once from a recorded female voice and once from their own voice. It is well known that repeated presentation of a stimulus results in the suppression of neural activity in regions that process that stimulus, a neural

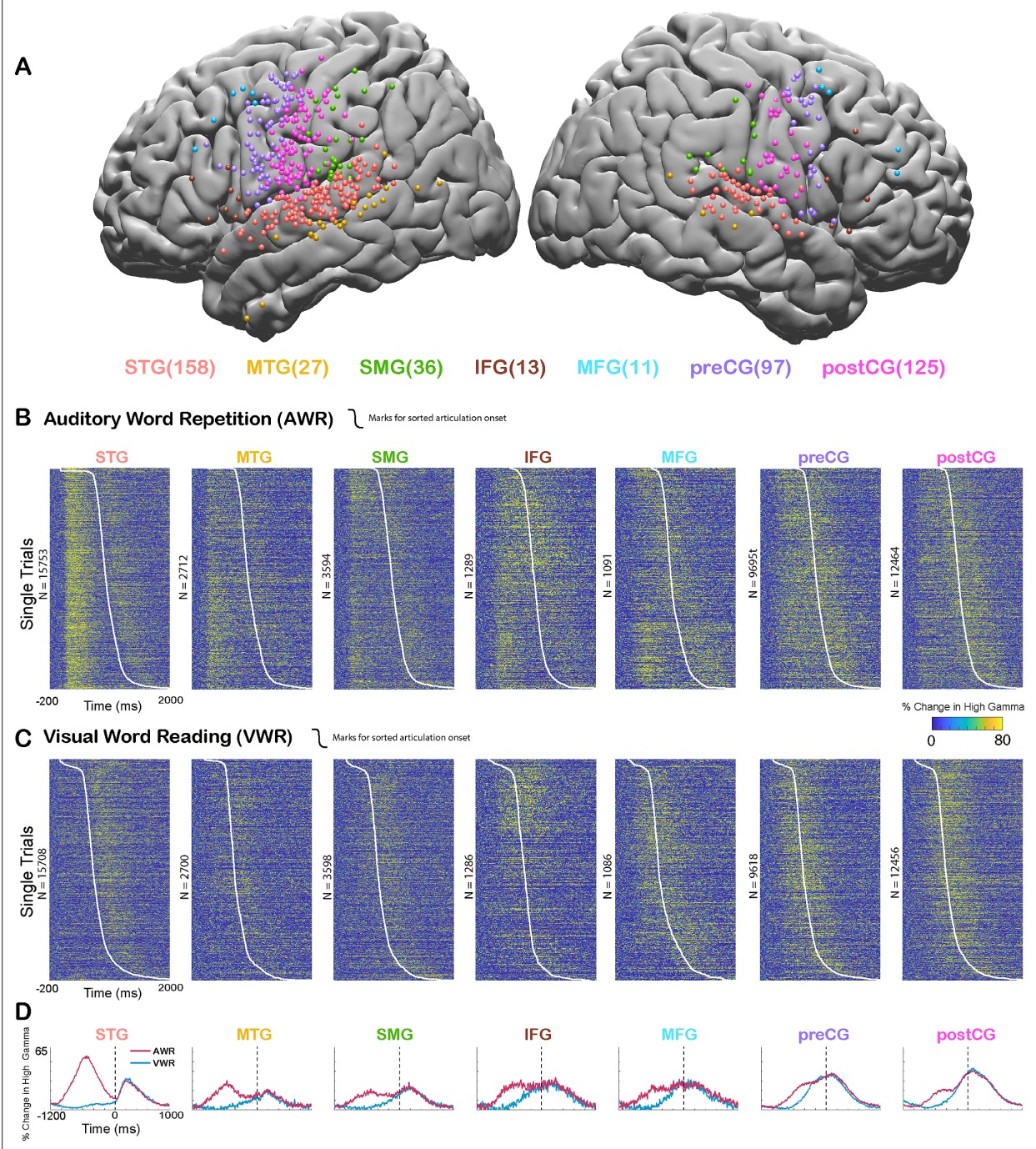

**Figure 1.** Cortical responses during speech tasks. (**A**) Electrodes from all participants (n=35) are shown on a template brain with different colors corresponding to different regions (number of electrodes in each region denoted in the parentheses). (**B**) High gamma broadband responses (70–150 Hz) for individual trials in an auditory word repetition task are shown for each region. (**C**) High gamma responses for individual trials in a visual word reading task are shown for each region. Trials are sorted with respect to speech onset (white line). (**D**) Mean high gamma broadband responses averaged across trials are shown for each region with the width representing the standard error of the mean across electrodes. Time = 0 indicates speech production onset.

adaptation phenomenon referred to as repetition suppression (*Grill-Spector et al., 2006*; *Todorovic and de Lange, 2012*). To ensure that our observed suppression of neural activity in auditory regions was not due to repetition suppression, but rather was induced by speech production, we performed a visual word reading (VWR) task, in which participants hear the auditory stimulus only once (from their

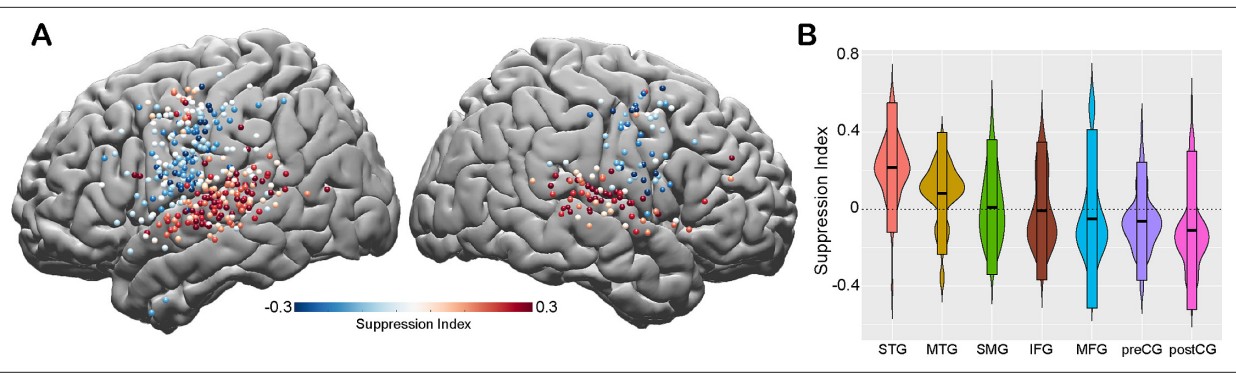

**Figure 2.** Spatial topography of speech-induced auditory suppression. (**A**) Suppression indices for all electrodes are shown on a template brain. Red color tones indicate smaller neural activity during speaking, while blue electrodes indicate larger neural activity during speaking compared to listening in the auditory word repetition task. (**B**) Suppression indices averaged across electrodes are shown for each region sorted from largest to smallest mean suppression index. Boxplots indicate mean ± SD.

own voice). Response magnitudes during speaking in the AWR and VWR tasks were similar (paired t-test: t(466)=0.62, p=0.53), characterized by a strong correlation across electrodes (Pearson's correlation: r=0.9006, p=0). These results suggested that repetition of the auditory stimulus in the AWR task did not affect response magnitudes and the observed reduction in response magnitudes was induced by speech production.

To quantify the amount of speech-induced suppression, we calculated a suppression index (SuppI) for each electrode by comparing neural responses during listening versus speaking in the AWR task (SuppI = Listen-Speak/Listen+Speak; *see Materials and methods*). A positive SuppI indicated a response suppression during speaking compared to listening and was observed most strongly in middle to posterior parts of STG, followed by MTG and SMG. A negative SuppI indicated a response enhancement during speaking compared to listening and was observed in motor regions, most strongly in the postCG (***Figure 2A and B***).

After mapping the topographical distribution of SuppI across the cortex, we focused on understanding the functional role of auditory suppression in speech monitoring. We hypothesized that the degree of speech-induced auditory suppression should be tightly linked to sensitivity to speech errors, as predicted by current models (***Houde and Nagarajan, 2011***; ***Tourville and Guenther, 2011***) and neural data in non-human primates (***Eliades and Wang, 2008***). To test this hypothesis, we used an additional task, in which we delayed the auditory feedback (DAF) during speech production to disrupt speech fluency. In this task, 14 participants repeated the VWR task while they were presented with their voice feedback through earphones either simultaneously (no-delay) or with a delay (50, 100, and 200 ms; *see Materials and methods*). In a previous study (***Ozker et al., 2022***), using the same dataset, we demonstrated that participants slowed down their speech in response to DAF (articulation duration; $DAF_0$: 0.698, $DAF_{50}$: 0.726, $DAF_{100}$: 0.737, and $DAF_{200}$: 0.749 ms). Moreover, auditory regions exhibited an enhanced response that varied as a function of feedback delay, likely representing an auditory error signal encoding the mismatch between the expected and the actual feedback. However, those results were not directly linked to auditory suppression.

Here, we compared neural responses in the AWR and the DAF tasks to test whether auditory regions that exhibit strong speech-induced suppression also exhibit large auditory error responses to DAF, which would indicate strong sensitivity to speech errors. In a single participant, we demonstrated that a representative electrode on the STG with strong auditory suppression (average % signal change in 0–500 ms; Listen: 124±7, Speak: 20±3, SuppI: 0.27) exhibited significant response enhancement ($DAF_0$: 135±12, $DAF_{50}$: 134±8, $DAF_{100}$: 175±10, $DAF_{200}$: 208±17, ANOVA: F(3, 116)=8.5, p=3.7e-05) (***Figure 3A and B***), while a nearby electrode with weaker auditory suppression (Listen: 116±6, Speak: 80±4, SuppI: 0.06) did not exhibit significant response enhancement with feedback delays ($DAF_0$: 360±29, $DAF_{50}$: 328±24, $DAF_{100}$: 379±31, $DAF_{200}$: 419±30, ANOVA: F(3, 116)=1.73, p=0.16) (***Figure 3C and D***).

To quantify the auditory error response and measure the sensitivity of a cortical region to DAF, we calculated a sensitivity index (SensI) for each electrode by correlating the delay condition and the

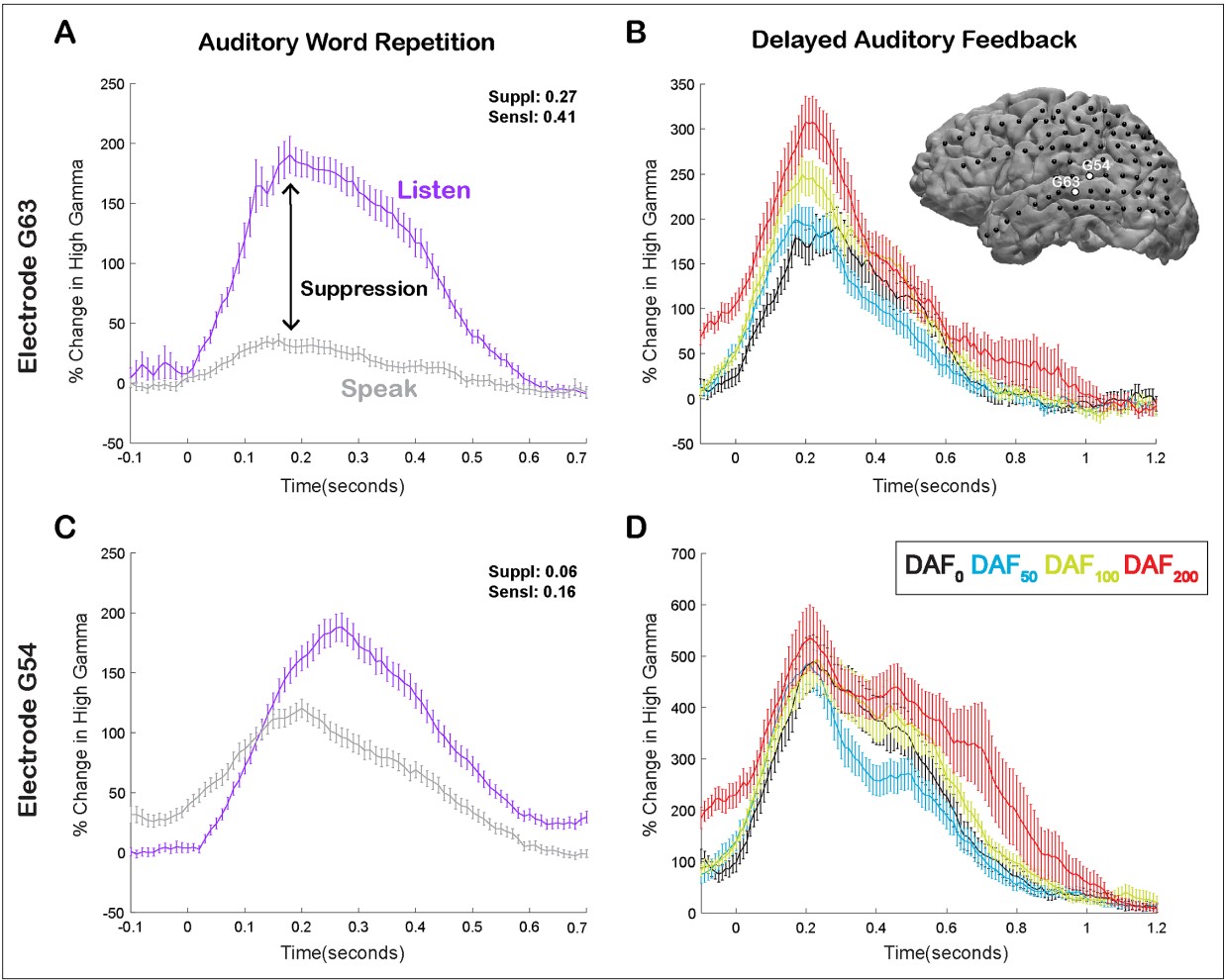

**Figure 3.** Speech-induced auditory suppression and sensitivity to delayed auditory feedback (DAF) in representative electrodes in a single participant. (**A**) High gamma broadband response (70–150 Hz) in electrode G63 showing a large amount of auditory suppression during speaking words compared to listening to the same words. Error bars indicate SEM over trials. (**B**) High gamma responses in electrode G63 to articulation of words with DAF. 0 s indicate the onset of the perceived auditory feedback. Inset figure shows the cortical surface model of the left hemisphere brain of a single participant. Black circles indicate the implanted electrodes. White highlighted electrodes are located on the middle (G63) and caudal (G54) superior temporal gyrus (STG). (**C**) High gamma response in electrode G54 showing a small degree of auditory suppression during speaking words compared to listening. (**D**) High gamma response in electrode G54 locked to articulation of words during DAF. 0 s indicate the onset of the perceived auditory feedback.

average neural response across trials (*see Materials and methods*). A large SensI indicated a strong response enhancement (large auditory error response) with increasing delays. The degree of both speech-induced suppression and sensitivity to DAF were highly variable across the cortex, SuppI ranging from –0.46 to 0.53 and SensI ranging from –0.62 to 0.70. The largest SuppI and SensI as well as a strong overlap between the two measures were observed in the STG, suggesting that auditory electrodes that show speech-induced suppression are also sensitive to auditory feedback perturbations (*Figure 4A–C*). We validated this relationship by revealing a significant correlation between SuppI and SensI of auditory electrodes (n=57, Pearson's correlation: r=0.4006, p=0.002) supporting our hypothesis and providing evidence for a common neural mechanism (*Figure 4D*).

Our neural analysis revealed that response magnitudes in auditory cortex were much larger when participants heard their simultaneous voice feedback in a DAF paradigm compared with producing speech without any feedback (DAF$_0$: no-delay trials) (average % signal change in 0–500 ms; DAF$_0$: 113±14, VWR: 41±7, compare gray lines in *Figure 3A and C* with black lines in *Figure 3B and D*, respectively). We were interested in dissociating if these larger responses were merely an effect of perceiving voice feedback through earphones instead of air or rather were specific to our DAF design, likely due to increased attentional demands. Therefore, four participants performed an additional

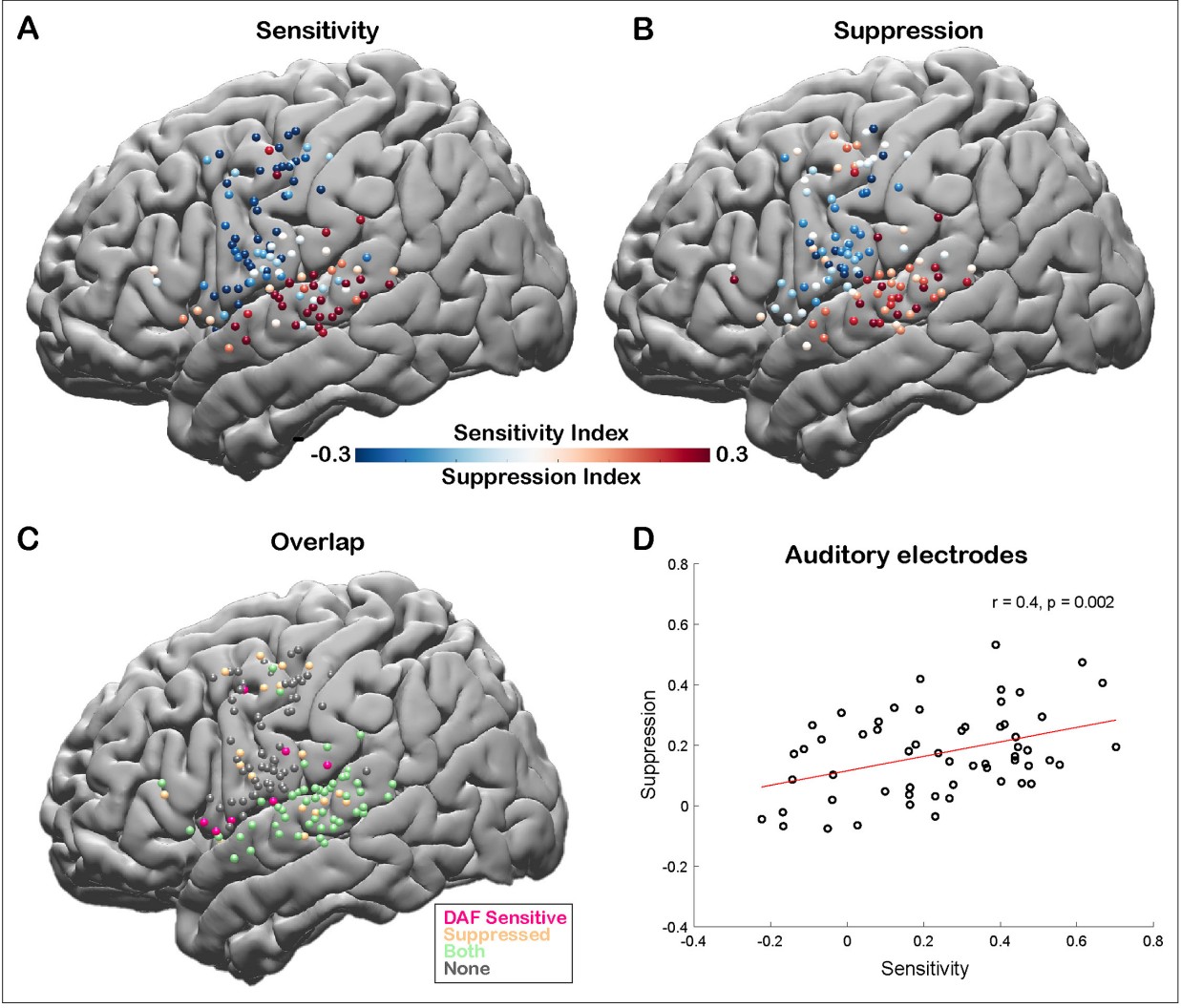

**Figure 4.** Correlation between speech-induced auditory suppression and sensitivity to delayed auditory feedback (DAF). (**A**) Sensitivity indices (SensI) for all electrodes are shown on a template brain (both right and left hemisphere electrodes were shown on the left hemisphere). Red tones indicate larger neural activity to increasing amount of delays in the DAF task, while blue tones indicate the opposite. (**B**) Suppression indices (SuppI) for all electrodes are shown on a template brain. Red tones indicate larger neural activity during listening compared to speaking in the auditory word repetition task, while blue tones indicate the opposite. (**C**) Electrodes that show either sensitivity to DAF (positive SensI value) or speech-induced auditory suppression (positive SuppI value), or both are shown on a template brain. (**D**) Scatter plot and fitted regression showing a significant correlation between sensitivity to DAF and speech-induced auditory suppression across auditory electrodes. Each circle represents an electrode's SensI and SuppI.

VWR task in which they were presented with their simultaneous voice feedback through earphones (VWR with auditory feedback [VWR-AF]). As previous studies have reported that DAF can increase voice intensity (*Yates, 1963*, *Howell and Archer, 1984*), we first verified whether participants spoke louder during the DAF task. A comparison of their voice intensity between $DAF_0$ (no-delay trials in the DAF task) and the VWR-AF (standard word reading with simultaneous feedback through earphones) conditions did not show a significant difference (voice intensity; $DAF_0$: 50±11 dB, VWR: 49±12 dB; paired t-test: t(118)=1.8, p=0.08). After verifying that the sound volume entering the auditory system is not statistically different in the two conditions, we compared the responses in the auditory cortex and found that overall response magnitudes were now on par across the two conditions ($DAF_0$: 89±17, VWR-AF: 82±17, *Figure 5A*). However, a detailed inspection of individual electrode responses revealed that some electrodes showed larger response to $DAF_0$, while others showed either larger responses to VWR-AF or similar responses to both conditions (*Figure 5B*). In a single participant, we demonstrated that adjacent electrodes in the STG that are only 5 mm apart exhibited completely

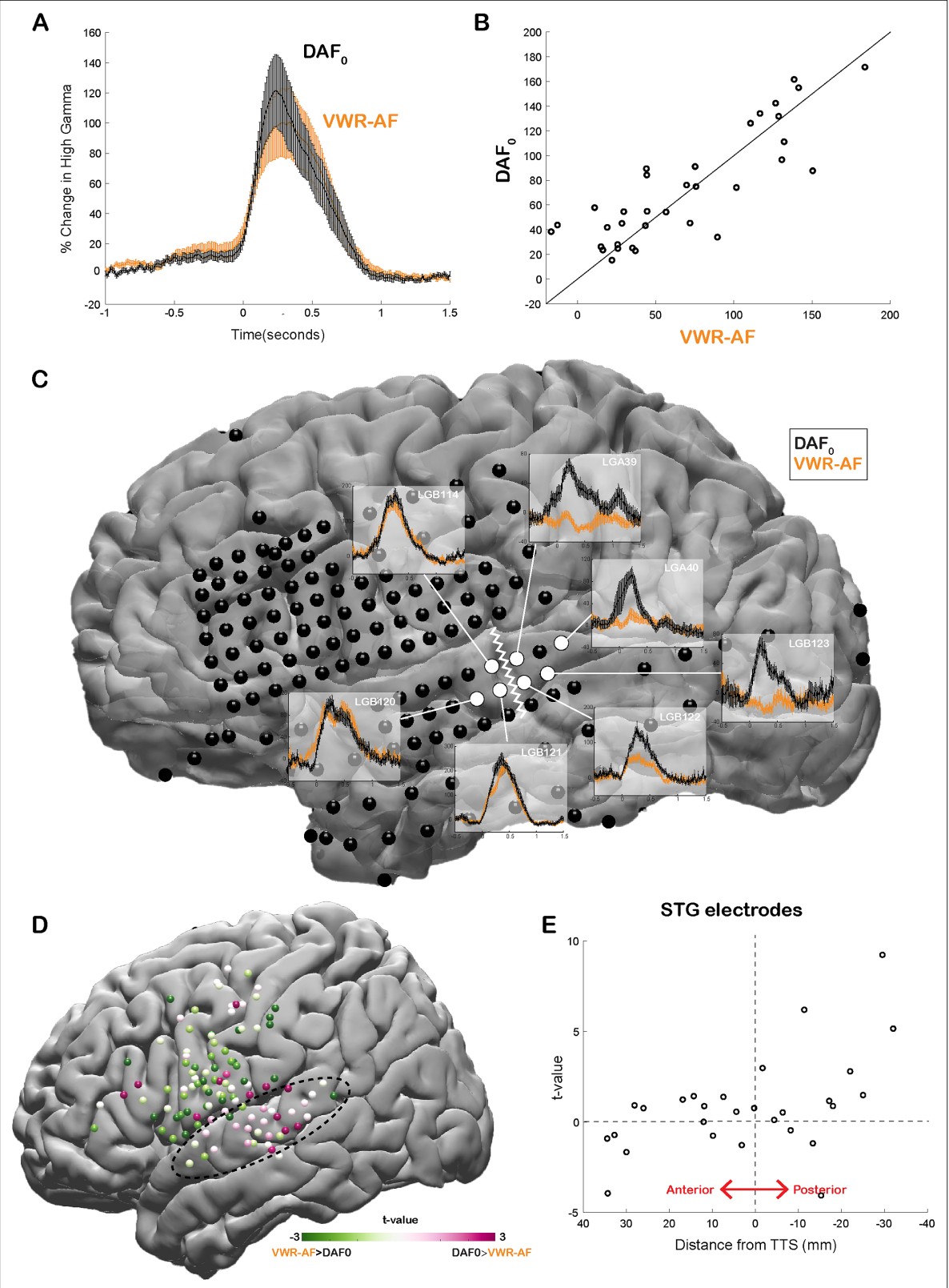

**Figure 5.** Effect of the delayed auditory feedback (DAF) paradigm on neural responses during speech. (**A**) High gamma broadband responses (70–150 Hz) averaged across auditory electrodes are similar during no-delay condition in the DAF task ($DAF_0$) and during visual word reading with auditory feedback (VWR-AF). Error bars indicate SEM across electrodes. (**B**) Scatter plot shows averaged high gamma responses (0–500 ms) for VWR-AF versus $DAF_0$ conditions for auditory electrodes. (**C**) High gamma responses for $DAF_0$ and VWR-AF are shown in representative auditory electrodes in a single

*Figure 5 continued*

participant. Electrodes that are posteriorly located on the superior temporal gyrus (STG) show larger responses to $DAF_0$ condition, while electrodes that are anteriorly located on the STG show similar responses to the two conditions. The lateral termination of the transverse temporal sulcus (TTS) is identified as a landmark (white zigzagged line) that separates the two different response patterns. (**D**) High gamma responses for $DAF_0$ and VWR conditions were compared and resulting t-values are shown for all electrodes on a template brain. Pink color tones indicate larger responses to $DAF_0$, while green color tones indicate larger responses to VWR condition. (**E**) t-values calculated by comparing responses to $DAF_0$ and VWR conditions are shown for all auditory electrodes with respect to their anterior-to-posterior positions to the TTS.

different response patterns. Electrodes in the more posterior parts of STG showed larger responses to $DAF_0$, while electrodes in more anterior parts showed similar responses to $DAF_0$ and VWR-AF (*Figure 5C*). To determine an anatomical landmark at which the reversal of response patterns occurred in the STG, we used the lateral termination of the transverse temporal sulcus (TTS) (*Greenlee et al., 2011*; *Nourski et al., 2016*) based on the individual FreeSurfer segmentation of the participant's preoperative MRI. Across participants, this landmark corresponded to y coordinate = –22±2.

Next, we compared the response patterns in the two conditions for all electrodes across participants by calculating a t-value for each electrode (unpaired t-test: average responses from –200 to 500 ms). We demonstrated that auditory regions in posterior STG showed larger responses to $DAF_0$ condition, while frontal motor regions showed larger responses to VWR-AF (*Figure 5D*). Lastly, we examined STG electrodes alone, sorted by their anterior-to-posterior positions with respect to the TTS. In line with the results from the single participant, electrodes that were located posteriorly within a 1 cm distance from this anatomical landmark showed significantly larger responses to the $DAF_0$ condition (*Figure 5E*). These results suggest that posterior STG is more activated when participants are engaged in a speech production task that requires increased effort and attention.

## Discussion

Our study provides a detailed topographical investigation of speech-induced auditory suppression in a large cohort of neurosurgical participants. We found that while the strongest auditory suppression was observed in the STG, the degree of suppression was highly variable across different recording sites. To explain this variability, we considered the functional role of auditory suppression in speech monitoring. We showed that delaying auditory feedback during speech production enhanced auditory responses in the STG. The degree of sensitivity to feedback delays was also variable across different recording sites. We found a significant correlation between speech-induced suppression and feedback sensitivity, providing evidence for a shared mechanism between auditory suppression and speech monitoring. While there was no anatomical organization for auditory suppression and feedback sensitivity in the STG, we found an anterior-posterior organization for the effect of attention on feedback sensitivity. Auditory sites that lie posterior to the lateral termination of the TTS in the STG showed stronger activation during the DAF task compared to a standard word reading task, even for trials in which participants received simultaneous feedback, demonstrating attentional modulation of feedback sensitivity.

We observed the strongest speech-induced suppression in the middle and posterior parts of the STG. In line with previous iEEG studies, we found that degree of suppression was variable across different recording sites in the STG without any anatomical organization (*Flinker et al., 2010*; *Greenlee et al., 2011*; *Nourski et al., 2016*). So far, a clear gradient for speech-induced suppression has never been reported in the STG but only in the Heschl's gyrus and superior temporal sulcus by studies that used comprehensive depth electrode coverage within the temporal lobe (*Nourski et al., 2016*; *Nourski et al., 2021*).

We found only a few sites with speech-induced enhancement and several sites with no response change. Based on single-unit recordings in non-human primates, it is known that majority of neurons in the non-core auditory cortex exhibits suppression, while a smaller group exhibits excitation during vocalization. It is difficult to isolate speech-induced enhancement in human studies, because measurements reflect the average response of the underlying neural population, which is dominated by suppressed responses. A previous non-human primate study suggested that there might be a division of labor between the suppressed and excited neurons. They showed that when an external auditory stimulus is presented concurrently during vocalization, neurons that showed vocalization-induced

suppression did not respond to the external stimulus. In contrary, neurons that showed vocalization-induced excitation responded even more when external stimulus is concurrently presented during vocalization, suggesting a role in maintaining sensitivity to the external acoustic environment (*Eliades and Wang, 2003*). In humans there might be a similar division of labor between auditory sites that were suppressed and non-suppressed, such that while suppressed sites are engaged in monitoring self-generated sounds, non-suppressed sites maintain sensitivity to external sounds. But unfortunately, our study did not include the necessary experimental conditions to directly test this hypothesis.

Our broad topographical search using subdural electrodes revealed additional sites outside the canonical auditory regions in the STG that showed speech-induced suppression, mainly in the MTG, and a few others in the SMG and preCG. Sensorimotor regions in the preCG including inferior frontal and premotor cortices are known to activate during passive listening tasks (*Wilson et al., 2004*; *Pulvermüller et al., 2006*; *Cogan et al., 2014*), and show tuning to different acoustic properties of speech similar to the auditory regions in the STG (*Mesgarani et al., 2014*; *Cheung et al., 2016*). Our results showed that isolated sites in these frontal motor regions were sensitive to DAF, confirming their auditory properties and suggesting their involvement in speech monitoring.

Current models of speech motor control predicted a shared mechanism between auditory suppression and sensitivity to speech errors, suggesting a role for auditory suppression in speech monitoring (*Houde and Nagarajan, 2011*; *Tourville and Guenther, 2011*). Behavioral evidence in human studies showed that when auditory feedback is delayed in real time, speakers attempt to reset or slow down their speech (*Lee, 1950*; *Fairbanks, 1955*; *Stuart et al., 2002*). Similarly, when fundamental frequency (pitch) or formant frequencies of the voice are shifted, speakers change their vocal output in the opposite direction of the shift to compensate for the spectral perturbation (*Houde and Jordan, 1998*; *Jones and Munhall, 2000*; *Niziolek and Guenther, 2013*). Neurosurgical recordings and neuroimaging studies that investigate the brain mechanism of auditory feedback processing demonstrated that these feedback-induced vocal adjustments are accompanied by enhanced neural responses in various auditory regions (*Tourville et al., 2008*; *Behroozmand et al., 2009*; *Behroozmand et al., 2015*; *Ozker et al., 2022*). However, it has not been clear whether it is the same or different neural populations that exhibit speech-induced suppression and enhanced responses to auditory feedback perturbations. Only in a non-human primate study, which recorded single-unit activity in auditory neurons of marmoset monkeys, it was shown that neurons that were suppressed during vocalization exhibited increased activity during frequency-shifted feedback (*Eliades and Wang, 2008*). In contrast, to replicate this finding in humans, a previous iEEG study by *Chang et al., 2013*, used frequency-shifted feedback during vowel production and found that most suppressed auditory sites did not overlap with those sensitive to feedback alterations. Using DAF instead of frequency-shifted feedback, we demonstrated a significant overlap of two neural populations in the STG, along with a strong correlation between the degree of speech-induced suppression and sensitivity to auditory feedback. This discrepancy may be due to different methods of calculating sensitivity to altered feedback. In our study, sensitivity was determined by comparing responses to delayed and non-delayed feedback during production, whereas Chang et al. compared perturbed feedback responses during production and listening. One possibility is that our approach identifies a larger auditory neural population in the STG sensitive to altered feedback. Alternatively, it could indicate a larger population highly sensitive to temporal rather than spectral perturbations in auditory feedback. Thus, we observe a wide overlap of the two neural populations in the STG showing both speech-induced suppression and sensitivity to auditory feedback. Replaying a recording of the participants' own delayed voice back to them, which we were unable to complete in this study, would have made the results of the two studies more comparable while also completely eliminating the possibility of a sensory explanation for the observed response enhancement.

Forward models of speech production suggest that a mismatch between the predicted and the actual auditory feedback is encoded by a response enhancement in the auditory cortex signifying an error signal (*Houde and Nagarajan, 2011*; *Tourville and Guenther, 2011*; *Hickok, 2012*). Our results suggested that attention to one's own speech stream during adverse speaking conditions, such as during an auditory feedback perturbations task, might also contribute to the response enhancement in the auditory cortex. Auditory feedback control of speech was thought to be involuntary and not subject to attentional control, because several previous studies showed that participants produced compensatory responses to pitch shifts even when they were told to ignore feedback perturbations

(*Munhall et al., 2009*; *Zarate et al., 2010*; *Keough et al., 2013*). However, prolonging pitch shift duration resulted in an early vocal response that opposes the pitch shift direction and a later vocal response that follows the pitch shift direction suggesting an interplay between reflexive and top-down processes in controlling voice pitch (*Hain et al., 2000*; *Burnett and Larson, 2002*). More recent EEG studies demonstrated that dividing attention between auditory feedback and additional visual stimuli or increasing the attentional load of the task affected vocal responses as well as the magnitude of ERP components, suggesting that attention modulates auditory feedback control on both a behavioral and a cortical level (*Tumber et al., 2014*; *Hu et al., 2015*; *Liu et al., 2015*; *Liu et al., 2018*). In our study, we found that neural responses in the posterior STG were larger for DAF$_0$ (randomly presented simultaneous feedback condition in the DAF task) as compared with the VWR-AF condition (consistent simultaneous feedback throughout standard word reading task), even though participants displayed similar vocal behavior in these two conditions. In light of the previous literature, we interpret these response differences as arising from an attentional load difference between the two tasks. In the DAF experiment, the auditory feedback was not consistent since no-delay trials were randomized with delay trials. This randomized structure of the paradigm with interleaved long delay trials (causing slowed speech) required conscious effort for speech monitoring and thus sustained attention. While remaining cautious about this interpretation and our study's limitation in attentional controls, we believe that this response enhancement represents an increased neural gain driven by attention to auditory feedback (*Hillyard et al., 1998*), and highlights the critical role of the posterior STG in auditory-motor integration during speech monitoring (*Hickok and Poeppel, 2000*), with its close proximity to the human ventral attention network comprising temporoparietal junction (*Vossel et al., 2014*). We leave it to future studies to include additional conditions to manipulate the direction and load of attention to further validate the influence of attention on speech monitoring.

## Materials and methods

### Participant information

The Institutional Review Board of NYU Grossman School of Medicine approved all experimental procedures. After consulting with the clinical-care provider, a research team member obtained written and oral consent from each participant. 35 neurosurgical epilepsy patients (19 females, mean age: 31, 23 left, 9 right, and 3 bilateral hemisphere coverage) implanted with subdural and depth electrodes provided informed consent to participate in the research protocol. Electrode implantation and location were guided solely by clinical requirements. Three patients were consented separately for higher density clinical grid implantation, which provided denser sampling of underlying cortex.

### iEEG recording

iEEG was recorded from implanted subdural platinum-iridium electrodes embedded in flexible silicon sheets (2.3 mm diameter exposed surface, 8×8 grid arrays, and 4–12 contact linear strips, 10 mm center-to-center spacing, Ad-Tech Medical Instrument, Racine, WI, USA) and penetrating depth electrodes (1.1 mm diameter, 5–10 mm center-to-center spacing 1×8 or 1×12 contacts, Ad-Tech Medical Instrument, Racine, WI, USA). Three participants consented to a research hybrid grid implanted which included 64 additional electrodes between the standard clinical contacts (16×8 grid with sixty-four 2 mm macro contacts at 8×8 orientation and sixty-four 1 mm micro contacts in between, providing 10 mm center-to-center spacing between macro contacts and 5 mm center-to-center spacing between micro/macro contacts, PMT Corporation, Chanhassen, MN, USA). Recordings were made using one of two amplifier types: NicoletOne amplifier (Natus Neurologics, Middleton, WI, USA), bandpass filtered from 0.16 to 250 Hz and digitized at 512 Hz. Neuroworks Quantum Amplifier (Natus Biomedical, Appleton, WI, USA) recorded at 2048 Hz, bandpass filtered at 0.01–682.67 Hz and then downsampled to 512 Hz. A two-contact subdural strip facing toward the skull near the craniotomy site was used as a reference for recording and a similar two-contact strip screwed to the skull was used for the instrument ground. iEEG and experimental signals (trigger pulses that mark the appearance of visual stimuli on the screen, microphone signal from speech recordings and feedback voice signal) were acquired simultaneously by the EEG amplifier in order to provide a fully synchronized dataset.

## Experimental design

### Experiment 1: AWR

35 participants performed the experiment. Stimuli consisted of 50 items (nouns) taken from the revised Snodgrass and Vanderwart object pictorial set (e.g. 'drum', 'hat', 'pencil') (*Rossion and Pourtois, 2004*; *Shum et al., 2020*). Auditory words presented randomly (two repetitions) through speakers. Participants were instructed to listen to the presented words and repeat them out loud at each trial.

### Experiment 2: VWR

The same 35 participants performed the experiment. Stimuli consisted of the same 50 words used in Experiment 1, however visually presented as text stimuli on the screen in a random order (two repetitions). Participants were instructed to read the presented word out loud at each trial.

### Experiment 3: DAF

A subgroup of 14 participants performed this experiment. Stimuli consisted of 10 different three-syllable words visually presented as text stimuli on the screen (e.g. 'envelope', 'umbrella', 'violin'). Participants were instructed to read the presented word out loud at each trial. As participants spoke, their voices were recorded using the laptop's internal microphone, delayed at four different amounts (no-delay, 50, 100, 200 ms) using custom script MATLAB, Psychtoolbox-3, available in GitHub (copy archived at *Ozker, 2024*) and played back to them through earphones. Trials, which consisted of different stimulus-delay combinations, were presented randomly (three to eight repetitions). Behavioral and neural data from the DAF experiment were used in a previous publication from our group (*Ozker et al., 2022*).

### Experiment 4: VWR-AF

A subgroup of four participants performed an additional VWR experiment, in which they were presented with the word stimuli as in Experiment 3 and heard their simultaneous (no-delay) voice feedback through earphones.

## Statistical analysis

Electrodes were examined for speech-related activity defined as significant high gamma broadband responses. Unpaired t-tests were performed to compare responses to a baseline for each electrode and multiple comparisons were corrected using the false discovery rate method (q=0.05). Electrodes that showed significant response increase ($p < 10^{-4}$) either before (−0.5 to 0 s) or after speech onset (0–0.5 s) with respect to a baseline period (−1 to –0.6 s) and at the same time had a large signal-to-noise ratio ($\mu/\sigma > 0.7$) during either of these time windows were selected. Electrode selection was first performed for each task separately, then electrodes that were commonly selected were further analyzed. For the analysis of the DAF experiment, one-way ANOVA was calculated using the average neural response as a dependent variable and feedback delay as a factor to assess the statistical significance of response enhancement in a single electrode.

## Experimental setup

Participants were tested while resting in their hospital bed in the epilepsy-monitoring unit. Visual stimuli were presented on a laptop screen positioned at a comfortable distance from the participant. Auditory stimuli were presented through speakers in the AWR and VWR experiments and through earphones (Bed Phones On-Ear Sleep Headphones Generation 3) in the DAF and in the VWR-AF experiment. Participants were instructed to speak at a normal voice level and sidetone volume was adjusted to a comfortable level at the beginning of the DAF experiment. DAF and VWR-AF experiments were performed consecutively and sidetone volume was kept the same in the two experiments. Participants' voice was recorded using an external microphone (Zoom H1 Handy Recorder). A TTL pulse marking the onset of a stimulus, the microphone signal (what the participant spoke), and the feedback voice signal (what the participant heard) were fed into the EEG amplifier as an auxiliary input in order to acquire them in sync with EEG samples. Sound files recorded by the external microphone were used for voice intensity analysis. Average voice intensity for each trial was calculated in dB using the 'Intensity' object in Praat software (*Boersma, 2001*).

## Electrode localization

Electrode localization in individual space as well as MNI space was based on co-registering a preoperative (no electrodes) and postoperative (with electrodes) structural MRI (in some cases a postoperative CT was employed depending on clinical requirements) using a rigid-body transformation. Electrodes were then projected to the surface of cortex (preoperative segmented surface) to correct for edema-induced shifts following previous procedures (*Yang et al., 2012*) registration to MNI space was based on a nonlinear DARTEL algorithm (*Ashburner, 2007*). Within participant anatomical locations of electrodes were based on the automated FreeSurfer segmentation of the participant's preoperative MRI. We recorded from a total of 3591 subdural and 1361 depth electrode contacts in 35 participants. Subdural electrode coverage extended over lateral temporal, frontal, parietal, and lateral occipital cortices. Depth electrodes covered additional regions to a limited extent including the transverse temporal gyrus, insula, and fusiform gyrus. Contacts that were localized to the cortical white matter were excluded from the analysis. To categorize electrodes in the STG into anterior and posterior groups, lateral termination of the TTS was used as an anatomical landmark (*Greenlee et al., 2011*; *Nourski et al., 2016*).

## Neural data analysis

Electrodes with epileptiform activity or artifacts caused by line noise, poor contact with cortex, and high-amplitude shifts were removed from further analysis. A common average reference was calculated by subtracting the average signal across all electrodes from each individual electrode's signal (after rejection of electrodes with artifacts). The analysis of the electrophysiological signals focused on changes in broadband high gamma activity (70–150 Hz). To quantify changes in the high gamma range, the data were bandpass filtered between 70 and 150 Hz, and then a Hilbert transform was applied to obtain the analytic amplitude.

Recordings from the DAF and VWR-AF experiments were analyzed using the multitaper technique, which yields a more sensitive estimate of the power spectrum with lower variance, thus is more beneficial when comparing neural responses to incremental changes in stimuli. Continuous data streams from each channel were epoched into trials (from –1.5 to 3.5 s with respect to speech onset). Line noise at 60, 120, and 180 Hz were filtered out. Three Slepian tapers were applied in timesteps of 10 ms and frequency steps of 5 Hz, using temporal smoothing (tw) of 200 ms and frequency smoothing (fw) of ±10 Hz. Tapered signals were then transformed to time-frequency space using discrete Fourier transform and power estimates from different tapers were combined (MATLAB, FieldTrip toolbox). The number of tapers (K) were determined by the Shannon number according to the formula: K=2*tw*fw-1 (*Percival and Walden, 1993*). The high gamma broadband response (70–150 Hz) at each time point following stimulus onset was measured as the percent signal change from baseline, with the baseline calculated over all trials in a time window from –500 to –100 ms before stimulus onset (data files containing high gamma activity recordings are available in GitHub).

## Suppl calculation

Suppression of neural activity is measured by comparing responses in two time periods in the AWR task. First time period was during listening the stimulus (0–0.5 s) and the second time period was during speaking (0–0.5 s). For each trial, average responses over Listen and Speak periods were found and suppression was measured by calculating Listen-Speak/Listen+Speak. Then suppression values were averaged across trials to calculate a single Suppl for each electrode. For the neural activity, raw high gamma broadband signal power was used instead of the percent signal change to ensure that the Suppl values varied between –1 and 1, indicating a range from complete enhancement to complete suppression respectively.

## Sensl calculation

Sensitivity to DAF is measured by comparing neural responses to increasing amounts of feedback delay. Neural responses in each trial were averaged in a time period following the voice feedback (0–0.5 s). For each electrode, a Sensl was calculated by measuring the trial-by-trial Spearman correlation between the delay condition and the averaged neural response. A large sensitivity value indicated a strong response enhancement with increasing delays.

## Acknowledgements

This study was supported by grants from the NIH (F32 DC018200 to MO and R01NS109367, R01DC018805, R01NS115929 to AF) and the NSF (CRCNS 1912286 to AF) and by the Leon Levy Foundation Fellowship (to MO).

## Additional information

### Funding

| Funder | Grant reference number | Author |
|---|---|---|
| National Institute on Deafness and Other Communication Disorders | F32 DC018200 | Muge Ozker |
| National Institute of Neurological Disorders and Stroke | R01NS109367 | Adeen Flinker |
| National Institute on Deafness and Other Communication Disorders | R01DC018805 | Adeen Flinker |
| National Institute of Neurological Disorders and Stroke | R01NS115929 | Adeen Flinker |
| National Science Foundation | CRCNS 1912286 | Adeen Flinker |
| Leon Levy Foundation | Fellowship in Neuroscience | Muge Ozker |

The funders had no role in study design, data collection and interpretation, or the decision to submit the work for publication. Open access funding provided by Max Planck Society.

### Author contributions

Muge Ozker, Adeen Flinker, Conceptualization, Resources, Data curation, Software, Formal analysis, Supervision, Funding acquisition, Validation, Investigation, Visualization, Methodology, Writing - original draft, Project administration, Writing - review and editing; Leyao Yu, Data curation, Software, Formal analysis, Validation, Investigation, Visualization, Methodology, Writing - original draft; Patricia Dugan, Resources, Data curation, Investigation, Project administration; Werner Doyle, Orrin Devinsky, Resources; Daniel Friedman, Resources, Data curation, Investigation, Methodology

### Author ORCIDs

Muge Ozker https://orcid.org/0000-0001-7472-4528
Adeen Flinker https://orcid.org/0000-0003-1247-1283

### Ethics

The study was approved by the NYU Grossman School of Medicine Institutional Review Board (approved protocol s14-02101) which operates under NYU Langone Health Human Research Protections. Research studies are performed in accordance with the Department of Health and Human Services policies and regulations at 45 CFR 46. Before obtaining consent, all participants were confirmed to have the cognitive capacity to provide informed consent by a clinical staff member. Participants provided oral and written informed consent before beginning study procedures. They were informed that participation was strictly voluntary, and would not impact their clinical care. Participants were informed that they were free to withdraw participation in the study at any time. All study procedures were conducted in accordance with the Declaration of Helsinki.

Reviewer #1 (Public Review): https://doi.org/10.7554/eLife.94198.3.sa1
Reviewer #2 (Public Review): https://doi.org/10.7554/eLife.94198.3.sa2
Author response https://doi.org/10.7554/eLife.94198.3.sa3

# Additional files

## Supplementary files
• MDAR checklist

## Data availability
Data and code are available in GitHub (copy archived at *Ozker, 2024*).

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
