## [Editor Report · eLife assessment]

The manuscript describes human intracranial neural recordings in the auditory cortex during speech production, showing that the effects of delayed auditory feedback correlate with the degree of underlying speech-induced suppression. This is an **important** finding, as previous work has suggested that speech suppression and feedback sensitivity often do not co-localize and may be distinct processes, in contrast with findings in non-human primates where there is a strong correlation. The strength of the evidence is **convincing**, with appropriate experimental methods, data, and analysis.

---

## [Referee Report · Reviewer #1 (Public Review)]

Summary:

The manuscript describes a series of experiments using human intracranial neural recordings designed to evaluate processing of self-generated speech in the setting of feedback delays. Specifically, the authors aim to address the question about the relationship between speech-induced suppression and feedback sensitivity in the auditory cortex, which, relationship has been conflicting in the literature. They found a correlation between speech suppression and feedback delay sensitivity, suggesting a common process. Additional controls were done for possible forward suppression/adaptation, as well as controlling for other confounds due to amplification, etc.

Strengths:

The primary strength of the manuscript is the use of human intracranial recording, which is a valuable resource and gives better spatial and temporal resolution than many other approaches. The use of delayed auditory feedback is also novel and has seen less attention than other forms of shifted feedback during vocalization. Analyses are robust and include demonstrating a scaling of neural activity with the degree of feedback delay, more robust evidence for error encoding than simply using a single feedback perturbation.

Weaknesses:

Some of the analyses performed differ from those used in past work, which limits the ability to directly compare the results. Notably, past work has compared feedback effects between production and listening, which was not done here. There were also some unusual effects in the data, such as increased activity with no feedback delay when wearing headphones, that the authors attempted to control for with additional experiments, but remain unclear. Confounds by behavioral results of delayed feedback are also unclear.

Overall the work is well done and clearly explained. The manuscript addresses an area of some controversy and does so in a rigorous fashion, namely the correlation between speech-induced suppression and feedback sensitivity (or lack thereof). While the data presented overlap that collected and used for a previous paper, this is expected given the rare commodity these neural recordings represent. Contrasting these results to previous ones using pitch-shifted feedback should spawn additional discussion and research, including verification of the previous finding, looking at how the brain encodes feedback during speech over multiple acoustic dimensions, and how this information can be used in speech motor control.

---

## [Referee Report · Reviewer #2 (Public Review)]

Summary:

In "Speech-induced suppression and vocal feedback sensitivity in human cortex", Ozker and colleagues use intracranial EEG to understand audiomotor feedback during speech production using a speech production and delayed auditory feedback task. The purpose of the paper is to understand where and how speaker induced suppression occurs, and whether this suppression might be related to feedback monitoring. First, they identified sites that showed auditory suppression during speech production using a single word auditory repetition task and a visual reading task, then observed whether and how these electrodes show sensitivity to auditory feedback using a DAF paradigm. The stimuli were single words played auditorily or shown visually and repeated or read aloud by the participant. Neural data were recorded from regular- and high-density grids from the left and right hemisphere. The main findings were:

• Speaker induced suppression is strongest in the STG and MTG, and enhancement is generally seen in frontal/motor areas except for small regions of interest in dorsal sensorimotor cortex and IFG, which can also show suppression.

• Delayed auditory feedback, even when simultaneous, induces larger response amplitudes compared to the typical auditory word repetition and visual reading tasks. The authors presume this may be due to effort and attention required to perform the DAF task.

• The degree of speaker induced suppression is correlated with sensitivity to delayed auditory feedback, and is strongest for ~200 ms of delayed auditory feedback.

• pSTG (behind TTS) is more strongly modulated by DAF than mid-anterior STG

Strengths:

Overall, I found the manuscript to be clear, the methodology and statistics to be solid, and the findings mostly quite robust. The large number of participants with high density coverage over both the left and right lateral hemispheres allows for a greater dissection of the topography of speaker induced suppression and changes due to audiomotor feedback. The tasks were well-designed and controlled for repetition suppression and other potential caveats.

Weaknesses:

I am happy with the changes the authors made in response to my first round of comments.

The authors addressed my comments relating to plotting relative to the onset of articulation in Figure 1 and also addressed whether the amount of suppression varies according to more interfering delayed auditory feedback (though the correlations between sensitivity and suppression are a little noisy, they are positive). Finally, I am also satisfied with the inclusion of more group data in Figure 4.

---

## [Author Response]

The following is the authors’ response to the original reviews.

**Public Reviews:**

**Reviewer #1 (Public Review):**
Summary:The manuscript describes a series of experiments using human intracranial neural recordings designed to evaluate the processing of self-generated speech in the setting of feedback delays. Specifically, the authors aim to address the question about the relationship between speech-induced suppression and feedback sensitivity in the auditory cortex, whose relationship has been conflicting in the literature. They found a correlation between speech suppression and feedback delay sensitivity, suggesting a common process. Additional controls were done for possible forward suppression/adaptation, as well as controlling for other confounds due to amplification, etc.Strengths:The primary strength of the manuscript is the use of human intracranial recording, which is a valuable resource and gives better spatial and temporal resolution than many other approaches. The use of delayed auditory feedback is also novel and has seen less attention than other forms of shifted feedback during vocalization. Analyses are robust, and include demonstrating a scaling of neural activity with the degree of feedback delay, and more robust evidence for error encoding than simply using a single feedback perturbation.Weaknesses:Some of the analyses performed differ from those used in past work, which limits the ability to directly compare the results. Notably, past work has compared feedback effects between production and listening, which was not done here. There were also some unusual effects in the data, such as increased activity with no feedback delay when wearing headphones, that the authors attempted to control for with additional experiments, but remain unclear. Confounds by behavioral results of delayed feedback are also unclear.Overall the work is well done and clearly explained. The manuscript addresses an area of some controversy and does so in a rigorous fashion, namely the correlation between speech-induced suppression and feedback sensitivity (or lack thereof). While the data presented overlaps that collected and used for a previous paper, this is expected given the rare commodity these neural recordings represent. Contrasting these results to previous ones using pitch-shifted feedback should spawn additional discussion and research, including verification of the previous finding, looking at how the brain encodes feedback during speech over multiple acoustic dimensions, and how this information can be used in speech motor control.

We thank the reviewer for their comments and have addressed the concerns point by point in the section “Recommendation for Authors”.

**Reviewer #2 (Public Review):**
Summary:"Speech-induced suppression and vocal feedback sensitivity in human cortex", Ozker and colleagues use intracranial EEG to understand audiomotor feedback during speech production using a speech production and delayed auditory feedback task. The purpose of the paper is to understand where and how speaker-induced suppression occurs, and whether this suppression might be related to feedback monitoring. First, they identified sites that showed auditory suppression during speech production using a single-word auditory repetition task and a visual reading task, then observed whether and how these electrodes show sensitivity to auditory feedback using a DAF paradigm. The stimuli were single words played auditorily or shown visually and repeated or read aloud by the participant. Neural data were recorded from regular- and high-density grids from the left and right hemispheres. The main findings were:• Speaker-induced suppression is strongest in the STG and MTG, and enhancement is generally seen in frontal/motor areas except for small regions of interest in the dorsal sensorimotor cortex and IFG, which can also show suppression.• Delayed auditory feedback, even when simultaneous, induces larger response amplitudes compared to the typical auditory word repetition and visual reading tasks. The authors presume this may be due to the effort and attention required to perform the DAF task.• The degree of speaker-induced suppression is correlated with sensitivity to delayed auditory feedback. • pSTG (behind TTS) is more strongly modulated by DAF than mid-anterior STGStrengths:Overall, I found the manuscript to be clear, the methodology and statistics to be solid, and the findings mostly quite robust. The large number of participants with high-density coverage over both the left and right lateral hemispheres allows for a greater dissection of the topography of speaker-induced suppression and changes due to audiomotor feedback. The tasks were well-designed and controlled for repetition suppression and other potential caveats.Weaknesses:(1) In Figure 1D, it would make more sense to align the results to the onset of articulation rather than the onset of the auditory or visual cue, since the point is to show that the responses during articulation are relatively similar. In this form, the more obvious difference is that there is an auditory response to the auditory stimulus, and none to the visual, which is expected, but not what I think the authors want to convey.

We agree with the reviewer. We have updated Figure 1 accordingly.

(2) The DAF paradigm includes playing auditory feedback at 0, 50, 100, and 200 ms lag, and it is expected that some of these lags are more likely to induce dysfluencies than others. It would be helpful to include some analysis of whether the degree of suppression or enhancement varies by performance on the task, since some participants may find some lags more interfering than others.

We thank the reviewer for this suggestion. In the original analysis, we calculated a Sensitivity Index for each electrode by correlating the high gamma response with the delay condition across trials. To address the reviewer’s question, we now compared delay conditions in pairs (DAF0 vs DAF50, DAF0 vs DAF100, DAF0 vs DAF200, DAF50 vs DAF100, DAF50 vs DAF200 and DAF100 vs DAF200).

Similar to our Suppression Index calculation, where we compared neural response to listening and speaking conditions (Listen-Speak/Listen+Speak), we now calculated the Sensitivity Index by comparing neural response to two delay conditions as follows:

e.g. Sensitivity Index = (DAF50 – DAF0) / (DAF50 + DAF0). We used the raw high gamma broadband signal power instead of percent signal change to ensure that the Sensitivity Index values varied between -1 to 1.

As shown in the figure below, even when we break down the analysis by feedback delay, we still find a significant association between suppression and sensitivity (except for when we calculate sensitivity indices by comparing DAF50 and DAF100). Strongest correlation (Pearson’s correlation) was found when sensitivity indices were calculated by comparing DAF0 and DAF200.

As the reviewer suggested, participants found DAF200 more interfering than the others and slowed down their speech the most (Articulation duration; DAF0: 0.698, DAF50: 0.726, DAF100: 0.737, and DAF200: 0.749 milliseconds; Ozker, Doyle et al. 2022).

(3) Figure 3 shows data from only two electrodes from one patient. An analysis of how amplitude changes as a function of the lag across all of the participants who performed this task would be helpful to see how replicable these patterns of activity are across patients. Is sensitivity to DAF always seen as a change in amplitude, or are there ever changes in latency as well? The analysis in Figure 4 gets at which electrodes are sensitive to DAF but does not give a sense of whether the temporal profile is similar to those shown in Figure 3.

In Figure 4A, electrodes from all participants are color-coded to reflect the correlation between neural response amplitude and auditory feedback delay. A majority of auditory electrodes in the STG exhibit a positive correlation, indicating that response amplitude increases with increasing feedback delays. To demonstrate the replicability of the response patterns in Figure 3, here we show auditory responses averaged across 23 STG electrodes from 6 participants.

**Author response image 2. sa3fig2:** 

Response latency in auditory regions also increases with increasing auditory feedback delays. But this delayed auditory response to delayed auditory feedback is expected. In Figure 3, signals were aligned to the perceived auditory feedback onset, therefore we don’t see the latency differences. Below we replotted the same responses by aligning the signal to the onset of articulation. It is now clearer that responses are delayed as the auditory feedback delay increases. This is because participants start speaking at time=0, but they hear their voice with a lag so the response onset in these auditory regions are delayed.

According to models of speech production, when there is a mismatch between expected and perceived auditory feedback, the auditory cortex encodes this mismatch with an enhanced response, reflecting an error signal. Therefore, we referred to changes in response amplitude as a measure of sensitivity to DAF.

(4) While the sensitivity index helps to show whether increasing amounts of feedback delay are correlated with increased response enhancement, it is not sensitive to nonlinear changes as a function of feedback delay, and it is not clear from Figure 3 or 4 whether such relationships exist. A deeper investigation into the response types observed during DAF would help to clarify whether this is truly a linear relationship, dependent on behavioral errors, or something else.

We compared responses to delay conditions in pairs in the analysis presented above (response #2). We hope these new results also clarifies this issue and address the reviewer’s concerns.

**Recommendations for the authors:**

**Reviewer #1 (Recommendations For The Authors):**
Major points:(1) While the correlation between SuppI and SensI is clear here (as opposed to Chang et al), it is unclear if this difference is a byproduct of how SensI was calculated (and not just different tasks). In that paper, the feedback sensitivity was calculated as a metric comparing feedback responses during production and listening, whereas here the SensI is a correlation coefficient during production only. If the data exists, it would be very helpful to also show an analysis similar to that used previously (i.e. comparing DAF effects in both production and playback, either in correlations or just the 200ms delay response). One could imagine that some differences are due to sensory properties, though it is certainly less clear what delay effects would be on listening compared to say pitch shift.

We thank the reviewer for pointing this out. Indeed, the calculation of SensI is different in the two studies. In Chang et al. study, SensI was calculated by comparing perturbed feedback responses during production and passive listening. This is a very meticulous approach as it controls for the acoustic properties of the auditory stimuli under both conditions.

In our study, we didn’t have a passive listening condition. This would require recording the participants’ voice as they were speaking with DAF and playing it back to them in a subsequent passive listening condition. Therefore, we can’t completely eliminate the possibility that some differences are due to sensory properties. However, to address the reviewer’s concern, we examined the voice recordings of 8 participants for acoustic differences. Specifically, we compared voice intensities for different auditory feedback delays (0,50,100 and 200ms) and found no significant differences (F=0, p=0.091).

We think that the difference with the Chang et al. study is an important point to emphasize, therefore we now added in the Discussion:

“In contrast, to replicate this finding in humans, a previous iEEG study by Chang et al. (Chang, Niziolek et al. 2013) used frequency-shifted feedback during vowel production and found that most suppressed auditory sites did not overlap with those sensitive to feedback alterations. Using DAF instead of frequency-shifted feedback, we demonstrated a significant overlap of two neural populations in the STG, along with a strong correlation between the degree of speech-induced suppression and sensitivity to auditory feedback. This discrepancy may be due to different methods of calculating sensitivity to altered feedback. In our study, sensitivity was determined by comparing responses to delayed and non-delayed feedback during production, whereas Chang et al. compared perturbed feedback responses during production and listening. One possibility is that our approach identifies a larger auditory neural population in the STG sensitive to altered feedback. Alternatively, it could indicate a larger population highly sensitive to temporal rather than spectral perturbations in auditory feedback. Thus, we observe a wide overlap of the two neural populations in the STG showing both speech-induced suppression and sensitivity to auditory feedback. Replaying a recording of the participants' own delayed voice back to them, which we were unable to complete in this study, would have made the results of the two studies more comparable while also completely eliminating the possibility of a sensory explanation for the observed response enhancement.”

(2) I am still a bit unclear on how Experiment 4 is different than the no-delay condition in Experiment 3. Please clarify. Also, to be clear, in Experiments 1+2 the subjects were not wearing any headphones and had no additional sidetone?

It is correct that participants were not wearing earphones in Experiments 1&2 (with no additional sidetone), and that they were wearing earphones in Experiments 3&4.

For the “no delay” condition in the DAF experiment (Experiment 3), participants were wearing earphones and reading words with simultaneous auditory feedback. So, this condition was equivalent to visual word reading (Experiment 2), except participants were wearing earphones. Yet, neural responses were much larger for the “no delay” condition in the DAF experiment compared to visual word reading.

We suspected that larger neural responses in the DAF experiment were caused by hearing auditory feedback through earphones. To test and control for this possibility, in a subset of participants, we ran an additional visual word reading experiment (Experiment 4) with earphones and used the same volume settings as in the DAF experiment. We found that response magnitudes were now similar in the two experiments (Experiment 3 and 4) and earphones (with the associated increased sound amplitude) were indeed the reason for larger neural responses. Thus, Experiment 4 differs from the no-delay condition in Experiment 3 only in the stimuli read aloud.

(3) In Figure 3, why is the DAF200 condition activity so much bigger than the other conditions, even prior to the DAF onset? I worry this might bias the rest of the response differences.

In Figure 3B and 3D, time=0 indicates the onset of the perceived auditory feedback. Below we replotted the responses in the same two electrodes but now time=0 indicates the onset of articulation. We see that the peaking time of the responses are delayed as the auditory feedback delay increases. This is because participants start speaking at time=0, but they hear their voice with a lag so the response onset in these auditory regions are delayed. However, like the reviewer pointed out, the response for the DAF200 condition in Electrode G54 is slightly larger even at the very beginning. We think that this small, early response might reflect a response to the bone-conducted auditory feedback, which might be more prominent for the DAF200 condition. Nevertheless, we still see that response amplitude increase with increasing feedback delays in Electrode 63.

(4) Figure 4C, are the labeled recording sites limited to those with significant DAF and/or suppression?

In Figure 4C, we show electrodes that had significant high-gamma broadband responses during all tasks. We write in the Methods: “Electrodes that showed significant response increase (p < 10−4) either before (−0.5 to 0 s) or after speech onset (0 to 0.5 s) with respect to a baseline period (−1 to −0.6 s) and at the same time had a large signal-to-noise ratio (μ/σ > 0.7) during either of these time windows were selected. Electrode selection was first performed for each task separately, then electrodes that were commonly selected were further analyzed.”

(5) Were there any analyses done to control for the effects of vocal changes on the DAF neural responses? The authors' previous paper did note a behavioral effect. This is probably not trivial, as we may not know the 'onset time' of the response, in contrast to pitch shift where it is more regular. If the timing is unknown, one thing that could be tried is to only look early in DAF responses (first 50ms say) to make sure the DAF effects hold.

DAF involves two different perturbations: the absence of feedback at speech onset and the introduction of delayed feedback during playback. The timing of the behavioral effect in response to these two perturbations remains unclear. Aligning the neural responses to the production onset and examining the first 50ms would only capture the response to the acoustic feedback for the no-delay condition within that time window. Conversely, aligning the responses to the playback onset might miss the onset of the behavioral effect, which likely starts earlier as a response to the lack of feedback. We acknowledge the reviewer's point that this is a limitation of the DAF paradigm, and the behavioral effect is not as straightforward as that of pitch perturbation. However, we believe there is no clear solution to this issue.

Minor points:(1) Figure 3, it might be nice to show the SuppI and SensI on the plots to give the reader a better sense of what those values look like.

We included SuppI and SensI values in the new version of Figure 3.

**Reviewer #2 (Recommendations For The Authors):**
Minor Comments:(1) In Figure 1, it is unclear whether the responses shown in B-D correspond to the ROIs shown in Figure A - I am guessing so, but the alignment of the labels makes this slightly unclear, so I suggest these be relabeled somehow for clarity.

This is fixed in the updated version of Figure 1.

(2) In Figure 1D the difference in colors between AWR and VWR is difficult to appreciate - I suggest using two contrasting colors.

This is fixed in the updated version of Figure 1.

(3) Please add y-axis labels for Fig 3B-D. (I believe these are % signal change, but it would be clearer if the label were included).

This is fixed in the updated version of Figure 3.

(4) Can the authors comment on whether the use of speakers for AWR and VWR versus earphones for DAF and VWF- AF may have had an influence on the increased response in this condition? If the AWR were rerun using the headphone setup, or if DAF with 0 ms feedback were run with no other trials including lags, would the large differences in response amplitude be observed?

Participants were not wearing earphones in Experiments 1&2, and that they were wearing earphones in Experiments 3&4.

For the “no delay” condition in the DAF experiment (Experiment 3), participants were wearing earphones and reading words with simultaneous auditory feedback. So, this condition was equivalent to VWR (Experiment 2), except participants were wearing earphones. Yet, neural responses were much larger for the “no delay” condition in the DAF experiment compared to VWR.

Supporting the reviewer’s concerns, we suspected that larger neural responses in the DAF experiment were caused by hearing auditory feedback through earphones. To test and control for this possibility, in a subset of participants, we ran the VWR-AF experiment (Experiment 4) with earphones and used the same volume settings as in the DAF experiment. We found that response magnitudes were now similar in the two experiments (Experiment 3 and 4) and earphones were indeed the reason for larger neural responses.

(5) No data or code were available, I did not see any statement about this nor any github link or OSF link to share their data and/or code.

Data is available in the Github repository: flinkerlab/Sensitivity-Suppression